# Insomnia Medication Use by University Students: A Systematic Review

**DOI:** 10.3390/pharmacy11060171

**Published:** 2023-10-27

**Authors:** Menghan Wang, Richard Cooper, Dan Green

**Affiliations:** 1School of Medicine and Population Health, University of Sheffield, Sheffield S1 4DA, UK; mwang50@sheffield.ac.uk; 2College of Health and Life Sciences, Aston University, Birmingham B4 7ET, UK; d.green3@aston.ac.uk

**Keywords:** insomnia, sleep medication, university/college students, prevalence, determinants, systematic review

## Abstract

Problematic sleep or insomnia has been a recognised issue for many individuals in society, and university students can be of particular concern due to unique academic pressures. A systematic review was designed to summarise the current evidence about the extent of insomnia medication used by university students and identify characteristics of those more willing to use medication to manage insomnia. Searches were undertaken using Psych INFO, PubMed, Embase, and Web of Science, resulting in 25 eligible studies across multiple countries between 1994 and 2020. The prevalence of sleep medication use by students varied widely, from 2% to 41.2%, with an average of 13.1%. Female gender, students experiencing poor sleep, smoking, drinking stimulant beverages, and undertaking fewer physical activities were associated with the use of insomnia medication. Insomnia medication use exists within university student populations but appears to vary considerably worldwide; identifying multiple population characteristics associated with such use would offer opportunities to identify and support those affected.

## 1. Introduction

Disruption of sleep, clinically referred to as insomnia, is recognised as having negative consequences for many individuals across society. From the patient’s perspective, insomnia may impact their physical and mental conditions such as daytime dysfunction, diabetes, hypertension, depression, anxiety, and feeling stressed [1,2,3,4]. Previous research has also revealed that the quality of life for those experiencing insomnia is reduced due to decreased working ability and social performance [5], which might increase the possibility of reduced productivity, and even work accidents, from the community perspective. In addition, insomnia is associated with an increasing economic impact, due to more frequent hospital visits, medication consumption, and the use of associated health services [6,7].

The prevalence of insomnia in the general population is known to vary considerably, and a previous systematic review found that the prevalence ranged from 6% to 48% in various countries, such as Germany and Italy [8]. Previous research in Canada [9] using a telephone survey of community-dwelling adults found that around 10% were diagnosed with insomnia. The previous literature has indicated various key factors that might contribute to adults’ insomnia; specifically, younger females, those with lower educational levels, and those with unstable jobs or low income have been reported to be more likely to suffer from insomnia [10]. Persistent insomnia occurred more frequently in older populations [11,12]. Individuals with poor physical and/or mental health were more likely to be diagnosed as having insomnia [13].

Specific subgroups of the general population have been the focus of particular attention. One such group is university students, where the prevalence of insomnia has been estimated between 9.4% and 38.2% across various countries [14]. For instance, a cross-sectional study was conducted to examine the prevalence of insomnia among university students in Helsinki, with 32.4% of participants reporting poor sleep quality [15]. The transition of studying and living patterns from school to university has been cited as a possible unique cause of insomnia among university students [16,17,18]. Akram et al. [17] noted that independent living and learning might induce insomnia for many university students. This, in turn, has been argued to then further impact on students’ university studies, as adequate sleep is important for maintaining concentration [19].

Two main categories of approaches to managing insomnia have been identified: pharmacotherapy, and psychological therapies. Pharmacotherapy can be differentiated into six main categories. First, benzodiazepines (such as temazepam) are one kind of prescribed hypnotic for managing insomnia; however, they have several adverse effects, including cognitive and psychomotor impairments, amnesia, and the development of tolerance [20,21,22]. Developed later, and intended to address several of the issues associated with benzodiazepines, Z-drugs (such as zolpidem and zopiclone) have been used to manage insomnia; again, these also have several side effects, including headache, daytime sedation, and bitter taste [23]. Of most concern, though, is the potential for both benzodiazepines and Z-drugs to lead to misuse, dependence, and addiction; therefore, clinical recommendations are to use such medications for short-term use only [24,25]. Third, neurohormone drugs such as melatonin and ramelteon influence circadian rhythms for promoting sleep [26]. Several common adverse effects of melatonin include abdominal pain, somnolence (sleepiness), headache, and palpitations [27]. Fourth, orexin receptor antagonists (such as suvorexant) restrain the arousal system by mitigating the function of orexin neurons to promote sleep instead of wakefulness, with several side effects, such as headache, nightmares, nausea, fatigue, dizziness, dry mouth, and upper respiratory and urinary tract infections [28,29]. Fifth, sedative antihistamines (including first-generation ones such as promethazine, diphenhydramine, and doxylamine) have been used particularly among over-the-counter medications to treat insomnia, but again they have similar side effects, such as daytime sedation and psychomotor impairments [26]. Sixth, herbal products (such as valerian) have been used to relieve mild nervous tension and difficulty falling asleep, with few side effects found for valerian apart from the next-day hangover when taking higher doses [26]. With respect to university students, the deleterious effects of these insomnia medications may have impacts on both physical and cognitive awareness, such as decreased attention and fatigue, which also has the possibility of influencing their academic performance.

As for psychological therapies, cognitive behavioural therapy for insomnia (termed CBT-I) often consists of several different therapeutic activities, including education (educating sleep hygiene), stimulus control (reducing bad habits), sleep restriction/compression (limiting sleep time), and relaxation (meditation activities) [30,31]. When it comes to effectiveness, CBT-I or CBT combined with pharmacotherapies seems to show more advantages in promoting falling asleep, increasing sleep duration, and improving insomnia compared with the pharmacotherapies alone [32].

The population-level prevalence of insomnia medication use has been reported from a number of sources, including prescription records and self-report surveys. In the United States (US), for example, a national survey found that from 2005 to 2010, approximately 4% of adults over the age of 20 years had taken prescription insomnia medication, with higher use being positively associated with increased age and education level [33]. Another US study of individuals with an insomnia diagnosis found that 19% of participants used insomnia medication, with more than two-thirds (69.4%) of these users continuing to take prescribed insomnia medication for more than one year [34]. However, relatively little is known about the subpopulation of university students; this is despite insomnia being a recognised issue in this population and the relative accessibility and cost-effectiveness of insomnia medications being a driver for their use [20]. Research has highlighted that university students take insomnia medication to promote regular sleep, with the aim of improving their quality of life and academic performance [35].

There is therefore a need to understand the patterns and extent of insomnia medication use in university student populations, so as to inform support and treatment policy and practice; these include prescribing and OTC sales, which may involve pharmacists and other healthcare professionals. A systematic review can help to summarise the existing evidence and provide a comprehensive overview of the current understanding of insomnia medication use by university students; however, we could find no existing review. This paper reports on a systematic review that aimed to explore the extent of insomnia medication use by university students, and to identify the potential determinants that led to the usage of insomnia medication.

## 2. Methods

This systematic review has been registered on the Prospective Register of Systematic Reviews (PROSPERO), registration number CRD42021252823. PRISMA Checklists have been attached as Appendix B.

### 2.1. Search Strategy

Utilising a systematic search approach, four databases (Psych INFO, PubMed, Embase, and Web of Science) were searched on 18 January 2021. Supported by one information specialist at the University of Sheffield, M.W. designed the search terms, while R.C. and D.G. conducted the final check. The following search terms were used: (insomnia OR sleep initiation and maintenance disorder* OR sleep problem* OR sleep disorder* OR dyssomnia OR sleep deprivation OR sleeplessness) AND (sleeping pill* OR pharmaceutical sleep aid* OR sleep inducer* OR sleep promoting agent* OR sleep* medication) AND (universit* AND student* OR college* OR higher education). No extra search limitations were added during the search process. More literature was identified from the reference lists of the studies identified from the search, in addition to those studies that cited the papers from the search (backward and forward citations). M.W. conducted the scope search by initially using the search terms to identify and decide which studies were excluded. M.W. completed the data extraction from the included studies, while R.C. and D.G. independently checked the data.

### 2.2. Eligibility Criteria

The inclusion criteria were as follows: (a) included university students; (b) explored the use of insomnia medication; (c) published in English; (d) full text available.

Following these criteria, the title and abstract were reviewed first for inclusion, with full-text review as necessary. For exploring the extent, the prevalence of insomnia medication use by university students was the main reference parameter in the final inclusion. In this review, data were not pooled or aggregated, and they were extracted from the published articles as reported.

### 2.3. Critical Appraisal

For the review, the AXIS tool [36] was used for appraising cross-sectional studies, and the CASP (Critical Appraisal Skills Programme) checklist was used for cohort studies [37]. AXIS allows the researcher to evaluate the quality of the corresponding sections (introduction, methods, results, and discussion) in each study, guided by several questions outlined in the appraisal tool. CASP predominantly focuses on three groups of appraisal questions around the results, including the validity of results, the expression and reliability, and the significance of the results. For both appraisal tools, the evaluator assigned “Yes”, “No”, or “Don’t know” for each question, which could provide the reviewers an opportunity to assess individual parts of the study qualitatively. By using the same criteria, each study can be assessed against the same elements. The completed appraisal results for the reviewed studies are presented in Appendix A. No study had a high risk of bias with the screening questions from these two critical appraisal tools, and all studies were considered for synthesis in the review. The appraisal results were completed by M.W. and checked independently by R.C. and D.G.

## 3. Results

A total of 657 records were initially identified (Figure 1). These were assessed for inclusion based on their title, abstract and, if necessary, a full-text review; 623 were excluded due to non-relevance based on the inclusion criteria. In these excluded papers, some of the full articles were not available, and while some papers explored university students’ sleep quality they did not mention their usage of insomnia medication. The remaining 34 papers were checked further, and after removing 16 duplicates, 1 literature review, and 6 papers not providing insomnia prevalence data, 11 papers were initially identified. Through backward and forward citations of these 11 papers, another 14 papers were also included. Therefore, 25 records were finally retained for the review. All information about the demographics and methodologies of each study is summarised in Table 1.

A total of 25 papers were included in this systematic review, published between 1994 and 2020. Of these, 25 were empirical studies, including 24 cross-sectional studies and 1 longitudinal study. The included studies involved multiple countries: nearly half (n = 11) were undertaken in the United States, with fewer studies being identified in other countries, such as Ethiopia (n = 2) and Jordan (n = 2); only a single study was included in each of the remaining countries: Brazil, China, Croatia, Lebanon, Malaysia, Nigeria, Palestine, Peru, South Africa, and Thailand. Almost all of the studies recruited more female students. Regarding the instruments, more than two-thirds of the studies (n = 18) used the PSQI (Pittsburgh Sleep Quality Index) and two surveys utilised the SQI (Sleep Quality Index) [51,52]. Another four studies [38,39,42,43] were undertaken via self-administered questionnaires based on the literature review, while one study [47] used a sleep diary. Although these tools were mainly used for examining sleep quality, they also involved a question on the use of insomnia medication; therefore, the prevalence of insomnia medication use could be calculated from the responses to this question. All studies performing statistical analysis used the score (PSQI/SQI) or point scale (self-administered questionnaires) as the measurement for insomnia medication use, and they correlated this with some factors of interest (such as gender, sleep performance, and lifestyle factors).

Across the included studies, the prevalence of using insomnia medication ranged from 2% [51] to 41.2% [44], with an average of 13.1%. Here, the prevalence referred to the percentage of those students who used insomnia medication (i.e., those responding “YES” to the question on the use of insomnia medication in those instruments). Almost all of the included studies did not mention which specific medication(s) the university students had used. Two studies highlighted that around 10% of the sample population took OTC medications [38,43], and another article demonstrated that 4.8% used prescription drugs and 2.0% used OTC medications [47].

The gender difference in the use of insomnia medication was discussed by ten studies, but the trend of their results was not consistent. Six studies [38,39,41,52,55,60] found that there tended to be more female students using insomnia medication than males, and four of them provided the specific prevalence rate between females and males [38,52,55,60] (Table 2). Through comparing the mean PSQI scores, Becker et al. [41] found that female students used insomnia medication more frequently. Another article presented the results of a Pearson’s chi-squared test (*p* < 0.0001) and stated that significantly more females used insomnia medication, but it did not provide the specific prevalences [39]. However, in four other studies, there were higher percentages of male students using insomnia medication [44,48,54,56]. Two studies found a statistically significant difference between genders in the use of insomnia medication [56,60].

Almost all of the identified studies examined the prevalence of insomnia medication use among general student populations; there were two studies that specifically recruited medical students and reported a prevalence of insomnia medication use of 22.4% [42] and 8.6% [53]. In the study of Alqudah et al. [42], the prevalence of using insomnia medication was compared between students majoring in different courses, including nursing, pharmacy, dentistry, applied medical sciences, medicine, and surgery. Within this study, the highest prevalence rate (29.9%) appeared among pharmacy students. Moreover, another study compared the mean frequency of insomnia medication use among students with different levels of academic achievement, finding that students obtaining a pass grade had the highest use, compared with students obtaining weak, good, very good, or excellent grades (*p* < 0.001) [40].

The use of insomnia medication was significantly correlated with several sleep performances in three studies [44,49,61] (Table 3). Assaad et al. [44] reported that students taking insomnia medication more than once per week had higher odds of poor sleep compared with those students never taking medications or taking them less than once per week. Quick et al. [61] found a statistically significant relationship (*p* = 0.009) between sleep duration and the use of insomnia medication, which indicated that students who had taken insomnia medication more than once in the past month were more likely to have a shorter sleep duration compared with those had not taken insomnia medication in the past month. Lund et al. [49] found a significant relationship between sleep quality and the use of insomnia medication (*p* < 0.001); specifically, around 30% of students with poor sleep quality used insomnia medication at least once a month, whereas only 5% of those with good sleep quality used insomnia medication.

The influence of a number of lifestyle behaviours (such as smoking, drinking central nervous system (CNS)-active and caffeine-containing beverages, physical activity, and eating competence) on taking insomnia medication was explored in several studies [40,43,45,46,48,54,55,60]. Four studies [40,54,55,60] suggested that, compared with non-smokers, students who smoked had higher odds of using insomnia medication (Table 4). In addition, Lohsoonthorn et al. [55] and Sanchez et al. [60] also included both former and current smokers and explored their relationships with insomnia medication use; both studies found that these subgroups had higher odds of using insomnia medication.

Table 5 presents the specific relationships between the use of insomnia medication and several kinds of drinking behaviours, such as CNS-active and caffeine-containing beverages. Drinking alcohol was discussed in four studies [43,54,55,60], and overall their findings suggested that students with higher consumption, or more frequent drinking, had higher odds of taking insomnia medication. In addition to the results from Sanchez et al. [60], another three studies found a significant association between drinking alcohol and using insomnia medication [43,54,55]. Goodhines et al. [43] measured nine levels of alcohol frequency (from 0 “non-drinking in the past two months” to 8 “drink every day”)—a different approach from other studies measuring alcohol consumption (<1, 1–19, ≥20 drinks per month) [54,55,60]. In addition to alcohol, the consumption of another CNS-active drink (i.e., stimulant beverages) was mentioned to be significantly associated with the use of insomnia medication in three studies [54,55,60]; specifically, students consuming stimulant beverages had higher odds of using insomnia medication compared with students who did not consume such beverages.

Four studies [48,54,55,60] reported that individuals were less likely to use insomnia medication if they had undertaken physical activity (Table 6). The use of insomnia medication was compared between students who reported undertaking physical activities and those who did not in three studies [54,55,60], with another study [48] categorising students further among those who undertook sufficient activity and those who did not. All of these studies reported that students with either insufficient or no physical activity had higher odds of using insomnia medication.

Quick et al. [45] concluded that high levels of “eating competence” (i.e., having a positive eating attitude and habits) were associated with lower reported use of insomnia medication (*p* = 0.037). One study [46] found that students suffering from asthma or allergies reported more insomnia medication use compared with those individuals without these conditions (*p* < 0.005). Two studies identified in the review investigated the correlation between body weight and insomnia medication consumption. Body mass index (BMI) was found not to be significantly associated with the use of insomnia medication (*p* = 0.838) in the study of Vargas, Flores, and Robles [50]. A similar finding (*p* = 0.97) was obtained by Suhaimi et al. [62].

## 4. Discussion

This systematic review identified and summarised existing evidence about university students’ use of insomnia medication, finding that less than 1 in 8 students (and an average of 13.1%) reported using insomnia medication. Of note was that the prevalence varied considerably, ranging from 2% to 41.2% between these populations of students, with the prevalence reported in about half of the included studies as lower than 10%. In the wider literature, several studies explored the general medication use (including insomnia medication) among university students, and the prevalence of insomnia medication use identified here was also lower than 10% [63,64,65]. Previous studies of insomnia medication use among the general population found prevalences between 11% and 31% [66,67,68], which is in line with the studies showing higher prevalences in this review. The slightly higher prevalence in this systematic review might be attributed to the study design and sampling; for example, Molzon et al. [46] reported a prevalence of more than one-quarter of students (26.9%), but of note was that two-thirds of the participants were intentionally recruited from students with chronic illness (such as asthma and allergies). Those with additional health conditions may be more likely to suffer from insomnia and, therefore, take medication. Again, linked to findings in the wider literature, students reporting an existing health condition were also more likely to report taking insomnia medication, which could be adapted to individuals in general. Individuals with mental or physical health conditions seemed to use insomnia medication more frequently [69,70,71,72].

As well as the influence of health conditions, this study reviewed several additional factors associated with insomnia medication use among university students. These included sleep quality and lifestyle behaviours; students with poor sleep quality or short sleep duration were found to report greater use of insomnia medication, as also found in research among the general population [69,73,74]. Students with lifestyle behaviours such as drinking CNS-active and caffeine-containing beverages, smoking, and undertaking less physical activity were found to have higher odds of insomnia medication use. The link between student insomnia medication use and some of these lifestyle behaviours challenges existing research among general populations where, for example, there was no significant correlation between drinking alcohol and using insomnia medication [75], as well as studies where smokers did not report using more sleep medication than non-smokers [76]. However, drinking alcohol and smoking could increase the risk of suffering from insomnia [75,76], and it is speculated that the possibility of using insomnia medication would increase when people have insomnia. There would be another concern with the interaction between alcohol and insomnia medications (such as benzodiazepines and phenobarbital), as alcohol can strengthen the sedative effects resulting in CNS impairment [77]. In the wider literature, e-cigarette smoking has been increasingly explored, and Brett et al. [78], for example, found that compared with nicotine cigarettes, e-cigarettes increased the chances of insomnia medication use; no similar studies involving e-cigarettes were identified in this review, despite e-cigarette use being highest among young adults—the age bracket that aligns with most university students [79]. In another study [80], sleep quality improvements due to physical activity were highlighted, but the effect of physical activity on insomnia medication use was not identified in this study.

This review also explored whether insomnia medication use was associated with demographic factors such as gender, BMI, and university course, but we found no clear trends. Studies involving more general populations have found that females were more likely to use insomnia medication than males, regardless of age [81,82], which contrasts with the mixed findings from different studies in this review. Furthermore, BMI was not identified as having a significant association with students’ insomnia medication use in this review, but the two studies that reported this [50,62] had a modest sample sizes (n = 240 and n = 515, respectively) and may not be representative of overall student populations. Of note was that few studies in this review explored the association between university subject and insomnia medication use; an exception was the reporting on medical students in two studies [42,53], and this focus may be related to a heightened concern and interest in such courses.

The PSQI was by far the most commonly used instrument to explore insomnia and associated medication use in this review, with nearly three-quarters (n = 18) of studies using it. The SQI was also identified in a minority of studies, as well as the use of more generic questionnaires and, occasionally, sleep diaries. The popularity of the PSQI is not unexpected, as previous studies have confirmed its reliability and validity [83]. Of note, however, was that of the 18 studies using the PSQI, only 7 reported insomnia medication use prevalences close to (within 5% either side) the average prevalence for all PSQI studies (8.5%, 8.6%, 8.8%, 11.3%, 13.8%, 15.8%, and 17.6%). In contrast, in the studies not using the PSQI (n = 7), three studies had insomnia medication use prevalences close to (within 5% either side) the average prevalence calculated from those seven studies (9.7%, 11.5%, 15%). Therefore, the use of the PSQI might not be the sole optimal measuring tool to estimate prevalence, and further measuring tools could explore more specific detection [84].

This systematic review had several limitations. Selection bias may have been introduced, as some studies might have be missed, such as non-English publications or studies only providing the PSQI score without the prevalence rate. Moreover, selection bias could also have been caused by subjective views, as only one reviewer conducted the scope search by using the search terms to identify and decide which studies were excluded. Most of the studies focused primarily on exploring university students’ sleep patterns, quality, or habits, as well as their associated determinants, with only two studies being identified that had a specific focus on insomnia medication [38,43]. By utilising the PSQI as the measurement tool, the prevalence of using insomnia medication could be examined, as could the associated factors. However, insomnia medication was not the main focus. Finally, some caution is needed in interpreting the findings, due to the variable quality of the included studies, with the main concerns being small sample sizes and convenience sampling (with the former being associated with a higher prevalence of insomnia medication use).

The above limitations also reflect research gaps. The limited articles indicate the limited knowledge about insomnia medication consumption among university students. In addition, six of the included studies lacked statistical analysis examining the association between the use of insomnia medication and identified relevant factors (i.e., gender, academic performance, sleep performances, and lifestyle behaviours), limiting the further conclusions that can be drawn about specific student groups. Due to the limited extent of this research field, the depth of investigation about insomnia medication is difficult to reflect with the information currently available, increasing doubts about the accuracy of the current literature. More comprehensive and better-designed studies are essential for a detailed and robust focus on insomnia medication use among university students.

## 5. Conclusions

This review identified sizeable insomnia medication use by university students internationally, with conflicting differences seen between genders. Students having poor sleep, smokers, those drinking CNS-active and caffeine-containing beverages, and those undertaking fewer physical activities had higher odds of insomnia medication use. This review highlights university students as a group who may be at particular risk of insomnia and insomnia medication use; this review can provide a useful alert for university and health staff to pay more attention to how students cope with their insomnia.

## 6. Future Directions

These findings have several implications for policy, practice, and future research. They offer insights for those who provide support for students, such as university and healthcare staff, including GPs (physicians), and others involved in university student welfare. GPs can play an important role in increasing recommendations about the use of non-pharmacological approaches for insomnia. By guiding students to use insomnia medication if essential, medication safety can be increased. Furthermore, this offers insights into those who may be more likely to require or use insomnia medication, such as those who smoke or have existing health conditions; this has implications for those involved in the prescription and supply of insomnia medication, such as doctors, nurses, and pharmacists. Pharmacists are frontline medical professionals who are relatively accessible to university students compared to some other health professionals and have the opportunity to increase students’ awareness about using insomnia medication safely. A recent review has suggested three main areas where pharmacists can help, including deprescribing insomnia medicines, as well as wider collaboration and education activities [85]. This review also offers suggestions for future research and, in particular, the need to understand more about students’ experiences of insomnia medication use, given the dearth of identified studies with this specific field. Finally, this review identified the popularity of the PSQI, but also raised concerns about its accuracy in its ability to provide detailed insights into the prevalence of insomnia and insomnia medication use; future research is warranted to develop more specialised data collection instruments.

## Figures and Tables

**Figure 1 pharmacy-11-00171-f001:**
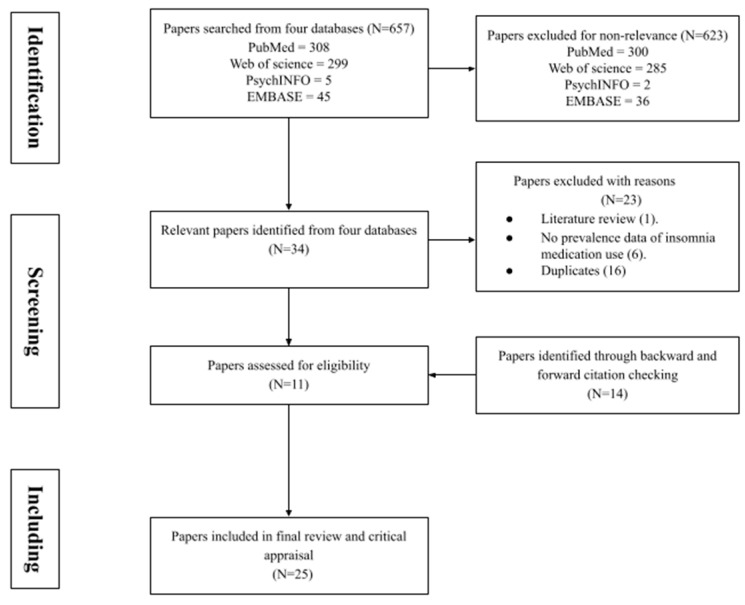
PRISMA diagram of the screening and inclusion process.

**Table 1 pharmacy-11-00171-t001:** The demographics and methodologies of the 25 reviewed studies.

Research	Year	Country	Research Design	Sample Size	Sex Female%	Mean Age (SD)	Research Instrument about Sleep	Prevalence of Using Insomnia Medication
Pillitteri et al. [38]	1994	US	Cross-sectional	278	66.2%	22.0 (6.3)	Self-administered questionnaires	9.7%
Baker et al. [39]	2008	South Africa	Cross-sectional	986	53.0%	Not Provide	Self-administered questionnaires	4.0%
Albqoor et al. [40]	2020	Jordan	Cross-sectional	1308	68.9%	Not Provide	PSQI	15.8%
Becker et al. [41]	2018	US	Cross-sectional	7626	70.1%	19.1 (1.4)	PSQI	24.4%
Alqudah et al. [42]	2019	Jordan	Cross-sectional	977	63.1%	20.9 (2.2)	Self-administered questionnaires	22.4%
Goodhines et al. [43]	2019	US	Longitudinal	171	67.8%	19 (1.4)	Self-administered questionnaires	15.0%
Assaad et al. [44]	2014	Lebanon	Cross-sectional	735	44.2%	20.6 (1.8)	PSQI	41.2%
Quick et al. [45]	2015	US	Cross-sectional	1035	61.0%	19.1 (1.1)	PSQI	11.3%
Molzon et al. [46]	2013	US	Cross-sectional	501	Not provided	19.4 (1.2)	PSQI	26.9%
Taylor et al. [47]	2010	US	Cross-sectional	1039	72.1%	20.4 (3.9)	Sleep diary	6.8%
Štefan et al. [48]	2018	Croatia	Cross-sectional	2100	50.0%	Not provided	PSQI	6.7%
Lund et al. [49]	2010	US	Cross-sectional	1125	62.7%	20.0 (1.3)	PSQI	17.6%
Vargas et al. [50]	2014	US	Cross-sectional	515	73.2%	21.7 (3.5)	PSQI	24.3%
Vail-Smith et al. [51]	2009	US	Cross-sectional	859	69.5%	Not provided	SQI	2.0%
Buboltz et al. [52]	2001	US	Cross-sectional	191	50.3%	19.0 (4.5)	SQI	11.5%
Correa et al. [53]	2017	Brazil	Cross-sectional	372	62.9%	Not provided	PSQI	8.6%
Lemma et al. [54]	2012b	Ethiopia	Cross-sectional	2230	22.4%	21.6 (1.7)	PSQI	8.5%
Lohsoonthorn et al. [55]	2013	Thailand	Cross-sectional	2854	67.4%	20.3 (1.3)	PSQI	6.2%
Sing et al. [56]	2010	China	Cross-sectional	529	54.6%	21.0 (1.8)	PSQI	13.8%
Sweileh et al. [57]	2011	Palestine	Cross-sectional	400	48.3%	20.2 (1.3)	PSQI	3.5%
Lemma et al. [58]	2012a	Ethiopia	Cross-sectional	2551	22.5%	Not provided	PSQI	8.8%
Seun-Fadipe et al. [59]	2017	Nigeria	Cross-sectional	505	49.5%	21.9 (2.7)	PSQI	20.4%
Sanchez et al. [60]	2013	Peru	Cross-sectional	2458	60.7%	20.9 (2.6)	PSQI	6.5%
Quick et al. [61]	2016	US	Cross-sectional	1252	58.9%	19.2 (3.5)	PSQI	5.8%
Suhaimi et al. [62]	2020	Malaysia	Cross-sectional	240	75.4%	21.2 (1.2)	PSQI	5.0%

**Table 2 pharmacy-11-00171-t002:** Studies covering the prevalence of insomnia medication use between genders.

Study	Female:Male (%)	*p*-Value
Buboltz et al. [52]	13.5:9.5	>0.05
Pillitteri et al. [38]	11.4:6.4	<0.21
Sanchez et al. [60]	7.5:5.1	0.012
Lohsoonthorn et al. [55]	6.2:6.0	0.614
Assaad et al. [44]	17.8:30.8	0.113
Štefan et al. [48]	5.9:7.6	0.231
Sing et al. [56]	8.4:20.4	<0.01
Lemma et al. [54]	7.4:8.8	0.356

**Table 3 pharmacy-11-00171-t003:** Studies covering the association between poor sleep performances and the use of insomnia medication.

Study	Sleep Performances	OR (95%CI)	*p*-Value
Assaad et al. [44]	Sleep quality	15.32 (4.89, 18.0)	-
Quick et al. [61]	Sleep duration	-	0.009
Lund et al. [49]	Sleep quality	-	<0.001

**Table 4 pharmacy-11-00171-t004:** Studies covering the association between smoking and using insomnia medication.

Study	OR (95%CI)	*p*-Value
Sanchez et al. [60]	2.11 (1.45–3.07)	<0.001
Lohsoonthorn et al. [55]	3.04 (1.42–6.54)	0.002
Lemma et al. [54]	2.84 (1.26, 6.43)	-
Albqoor et al. [40]	-	0.014

**Table 5 pharmacy-11-00171-t005:** Studies covering the association between drinking CNS-active and caffeine-containing beverages and using insomnia medication.

Study	Drinking Behaviour	OR (95%CI)	*p*-Value
Goodhines et al. [43]	Alcohol frequency	-	<0.01
Sanchez et al. [60]	Alcohol consumption	-	0.727
	Stimulant beverage consumption		0.005
Lohsoonthorn et al. [55]	Alcohol consumption	-	0.001
	Any stimulant beverage consumption	1.32 (0.76, 2.32)	-
Lemma et al. [54]	Alcohol consumption	-	0.003
	Khat consumption	-	<0.001
	Any caffeine-containing beverage consumption	1.15 (0.63, 2.13)	-

**Table 6 pharmacy-11-00171-t006:** Studies covering the association between decreasing physical activity and using insomnia medication.

Study	OR (95%CI)	*p*-Value
Sanchez et al. [60]	1.07 (0.76, 1.51)	-
Lohsoonthorn et al. [55]	0.53 (0.30, 0.93)	-
Štefan et al. [48]	3.80 (2.02, 7.13)	<0.001
Lemma et al. [54]	1.60 (0.88, 2.91)	-

## Data Availability

The data presented in this study are available in this article.

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
