# Peer review of "Insomnia Medication Use by University Students: A Systematic Review"

_pharmacy, 2023, doi:10.3390/pharmacy11060171_

Round 1

Reviewer 1 Report

Comments and Suggestions for Authors

Many thanks for giving me the opportunity to review this paper. The authors are addressing an important topic in health and education research, and the work is timely. There has been a lot of work to prepare this paper, and the standard of English is good in the introduction and methods although there are a few examples where the text is less clear and should be clarified. However the clarity of sentences in the results and discussion sections requires a full review by the authors as it was at times very difficult to follow.

One of the main issues here for the authors to clarify is whether their review was a systematic review or not. At present the paper does not have a focused research question, a PRISMA statement / flow diagram, or a dedicated study quality section which makes it read more like a scoping review. The authors also need to justify to the reader why they identified more than 50% of included studies from reference lists rather than the databases they searched, as some may interpret this as the search not being designed appropriately (for example, were the reference list papers not from the databases searched?) – this links to the decision to use the word ‘sleep’ in the medication related search terms, which I think may have created a search more narrow that it perhaps needed to be. There is also the question of whether they carried out independent extraction of data to improve quality.

Throughout the paper there could be much greater attention to detail, and clarification of points made. This includes key details for the methods section described below, as well as more information about included studies when they are mentioned and how the results of statistical tests are reported. Including an expanded ‘table 1’ describing key study attributes will help (based on Appendix A). The authors need also to add early on how the survey tools they mention like PSQI can help measure prevalence of medication use as this is not explained until the results/discussion.

There also needs to be more attention in the introduction to prior research in this field and adequate justification for this review being needed.

Please see below for a more detailed list of comments, organised by section.

Abstract

-          Please can the authors add greater clarity on what is meant by ‘more medication’

-          Please can the authors clarify what is meant by ‘with the gender difference’: is this meant to be that gender was taken into account as part of formal analyses? But the main results found mixed results regarding gender?

-          Please correct a typo: ‘…vary considerable internationally…’

Introduction

-          Please rephrase ‘…and even the whole society.’ To something like ‘and across society.’

-          Please rephrase ‘…due to more times of visiting hospitals…’ is this ‘more frequent hospital visits’?

-          Line 35-36. For the study referenced in Canada can the authors please specify what population was studied.

-          Line 36-41: This is quite a long sentence, can the authors please reword to be shorter and focus on the risk factors.

-          Line 60: Please can the authors rephrase ‘…but widely used for promoting…’ to ‘…have been seen to be used…’ as prescribing varies by country and region, and this prescribing is not considered in line with guidelines.

-          Line 63-65: can the authors please specify here which classes of medication they are referring to (i.e benzodiazepines and Z drugs)

-          Lines 73-82: Can the authors please strengthen the justification for this review by referring to existing evidence. For example are there previous literature reviews of the topic and what were their limitations? Are there individual studies and more recent papers meaning that a study bringing together fragmented evidence is needed?

-          In the introduction more broadly, can the authors also please summarise the importance of psychological therapies for insomnia, and the evidence base for students as this is of wider importance and helps to frame the subject in context

Methods

-          2.1: can the authors please clarify whether their systematic review was indeed a systematic review, as the line ‘systematic searching approach’ implies otherwise. You would also expect the review to be conducted in line with the PRISMA statement, can the authors please clarify whether this has been the case and if not ensure PRISMA statement is completed as an appendix and mentioned in the main text.

-          In the search strategy, it is noted that all medication related terms included the word ‘sleep’. Can the authors please comment on whether this may have restricted the search, and also for example by not mentioning specific drug classes or names. By having a group of terms specifically dedicated to insomnia condition, I don’t think adding ‘sleep’ added much and may actually have restricted the search unnecessarily. This may reflect why the final number of hits was <1000 across the databases.

-          Section 2.3. Can the authors please specify which author completed data extraction from the studies, and whether extraction was independently checked by other authors. In addition, for the appraisal please also specify who completed this in the text, and whether this was independently checked by other authors. Independent checking of data extraction and quality appraisal is an important quality marker for systematic reviews – I note that there is a statement in the discussion limitations section about one author ‘did the scope searching’ but I am not sure what this means, can the authors please clarify in the methods and discussion sections. I think it would strengthen the paper to have independent extraction of data on a proportion of papers, e.g. 20%.

-          ‘It has been suggested that no specific study had risks of bias with the screening questions of these two critical appraisal tools, and all studies could provide clear evidence for the main topic.’ Can the authors please clarify the meaning of this sentence in the text – were all studies rated as low risk of bias? How was this determined using the tools? The authors should include a full risk of bias assessment section in the results, so the reader can see for themselves how this influences the findings.

-          Can the authors please include in the methods section how they analysed and gathered and presented data on prevalence of medication use. How as prevalence defined, did the authors manipulate any data to make the rates similar across studies. Was this the only primary outcome they explored as this was not clear in the text and should be clarified. Also, the authors state that they aimed to ‘identify the potential determinants that led to the usage of insomnia medication,’ but there is no mention in the methods about what data they were extracting and analysing from included papers to address this aim. This must be clarified in the methods.

-          Can the authors please clarify why they included which instrument was used to measure insomnia symptoms in Appendix A and the main text. Why was this extracted and how was it helpful?

Results

-          Figure 1 is blurry, please can the authors amend to make clearer. It is usual convention for systematic reviews to include the number of citations excluded at the title, abstract and then full text stages. This includes any duplicates initially removed. Can the authors please update Figure 1 and the text in Section 3 to reflect this

-          Finding 14 papers for inclusion via backward and forward screening is more than 50% of the total. This could suggest that the initial search strategy did not pick up on relevant papers. Can the authors please review and briefly explain where these 14 papers came from (e.g. databases) in the main text, and how the review search did not identify them. If necessary, please update the limitations section to explain why the review search strategy did not identify them.

-          Typically in systematic review studies you would expect to see a large ‘Table 1’ which summarises key information from across all included studies. Please can the authors provide a Table 1 in the main text which does this for their 25 included studies. I can see in Appendix A that this is provided, but can this be included in the main text as table 1, and also to include further information such as what method was used to collect data (e.g. interview, survey, records), over how long the students were asked to report use of medication, who the students were and how they were recruited, and the numerator/denominator labels.

-           ‘Through checking backward and forward citations of these 11 papers, another 14 papers were also included. By excluding one literature review and six papers not providing the prevalence, 25 records were finally included into the review. All information about demographics and methodologies of each study was summarized in the Appendix A.’ Can the authors please revisit this sentence. I am unclear how 25 papers can be found when adding the backward/forward citations, with 7 then excluded, and 25 records were left for inclusion?

-          Line 132: please can the authors add a sentence clarifying what methods the papers used to collect their cross-sectional data. It looks like they were all surveys of some kind with different tools being used – can it be added to the main text how did these different tools capture medication use?

-          ‘Almost all the studies recruited more female students.’ Please can the authors provide figures in support of this in the text, e.g. mean + SD

-          ‘Regarding the instruments, …’ Please can the authors elaborate in the text what these instruments were used for

-          ‘Across the included studies, the prevalence of using insomnia medication ranged from 2% to 41.2%’ - please can the authors provide the references for the 2% and 41% figures in the text. Also, when referring to the prevalence it is important to include a denominator – is this for example the % of students who took part? Please clarify in the text

-          Lines 143 – 146: the comments on prevalence rates over time appear quite inconclusive, particularly given the differences in countries? Maybe looking at the USA data alone might help here.

-          Line 150: when discussing the prevalence between male and females, can the authors please note that the study method may influence the data. For example a study relying on survey data is dependent upon response characteristics whereas a study examining medical records is not subject to that bias, but may not capture over the counter pharmacy sales.

-          Lines 147, 154: please review and correct grammatical mistakes e.g. ‘… present the consistent results…’

-          Line 165: please can the authors specify the country of origin for these studies when describing them

-          Line 175: here the authors imply that statistical testing was used to correlate medication use with sleep quality. When reporting study findings like study 41, please can the authors make clear how they calculated the data (e.g. was it using a statistical test comparing prevalence values).

-          ‘Lund et al. [48] found insomnia medication use among students reporting good sleep quantity and sleep quality was around half that of those reporting poor quality sleep (p<0.001).’ Please can the authors revisit this sentence as I am not sure of the meaning

-          For Tables 2-5, and the main text where they are described, please can the authors be very clear on what they mean by ‘insomnia medication consumption’. Is this the proportion reporting that they take any medication, or a higher number of medications than others? When reporting OR and P values, please can the authors therefore also be clear on what these are referring to. For Table 5, this also needs to be more clearly defined in relation to ‘less insomnia medication’.

-          ‘several drinking consumptions’ please can the authors reword this as it is unclear currently

-          ‘One study [50] found that a higher response of using medication for insomnia was presented among the students suffering from asthma or allergies compared to those individuals without diseases (p<0.005).’ Can the authors please clarify in the text what is meant by ‘higher response of using medication’ as this is not clear currently.

-          ‘Almost all included studies did not mention which specific medication university students took. Merely, two studies both pointed that around 10% of sample population took OTC medications [35, 51], and another research result demonstrated that 4.8% used prescription drugs and 2.0% used OTC medications [52]’ This is a very important finding in the review, and should be placed early in the results section (it feels out of place where it is currently). It is very important that no studies actually recorded what medications students were using for insomnia.

Discussion

-          ‘…reported previous month use.’ Please can this be reworded as it is unclear currently

-          Lines 238-245, please can the authors review sentences to improve clarity and flow

-          Line 253: ‘…mental or physical health conditions…’

-          Line 255: ‘…this study…’

-          ‘In contrast, the effect of physical activity on insomnia medication use in this review did reflect wider literature which has highlighted sleep quality improvements due to physical activity [71].’ These appear to be two different points? This study is focused on medication use and study 71 is about physical activity and sleep quality. Please can the authors review this statement.

-          Lines 284-6: please can the authors remove the quote as use of the reference is enough

-          ‘Therefore, the use of the PSQI might not be the sole optimum measuring tool to estimate prevalence, and further measuring tools could explore more specific detection [75].’ Can the authors please clarify in the text how they reached this conclusion – the text immediately before implies that there was wide variation in reported use of these medications, so this might have nothing to do with the tools themselves and might be more to do with the country/area of study, time of conduct and student population included?

-          ‘By utilizing the PSQI as the measurement tool, the prevalence of using insomnia medication could be examined, as were the associated factors.’ I think that the authors need to add much more text to the introduction and methods to explain how such tools can help calculate these parameters.

-          ‘The limited articles have indicated the limited knowledge about insomnia medication consumption within university students, as some of existing literature also lacked statistical analysis of insomnia medication and its relevant factors.’ Can the authors please add more clarity in the text on what is meant by ‘relevant factors’ and how many studies did not report statistical analysis of this data.

-          In the conclusion it is said ‘also with the gender difference.’ However the findings were mixed with regards to gender? Can the authors please clarify what is meant here.

-          In the conclusion it is said ‘This systematic review has suggested that university students may suffer from health risk of taking medication…’ The study was not designed to assess this risk and did not report any data on risk, please can the authors remove this statement and perhaps instead focus on targets for future research and practice?

Comments on the Quality of English Language

See above main comments to authors

Author Response

Thanks for your comments and all amendments based on the comments have been highlighted in red in the revised file.

Reviewer 2 Report

Comments and Suggestions for Authors

Thank you for the opportunity to review the paper by Wang et al regarding "Insomnia medication use by university students: A systematic review." The topic of this systematic review is of interest to the health care and higher education communities. 

Strengths: The authors used appropriate methods to conduct this systematic review including registering with PROSPERO, reporting the specific search strategy for reproducibility, critical appraisal of each study included, and a PRISMA diagram showing the screening and inclusion process.

Deficiencies: The following citations were found and should be considered for inclusion in your systematic review.

Afshoon MA, Hosseini S, Hazar N, Vakili M, Rahmanian V. Zolpidem Use Among Dormitory Students in Yazd, Iran. International Journal of High Risk Behaviors and Addiction. 2020 Mar 31;9(1).

Afshoon MA, Hosseini S, Hazar N, Vakili M, Rahmanian V. Zolpidem Use Among Dormitory Students in Yazd, Iran. International Journal of High Risk Behaviors and Addiction. 2020 Mar 31;9(1).

Additional Questions: Regarding the characteristics of those students who are using more insomnia medication, did any students evaluate illicit drug use or nonmedical use of prescription drugs as a risk factor? Also, did any studies evaluate working outside of their academic careers or students who have children as a factor for increased use? Did any studies evaluate undergrad versus graduate studies? 

Comments on the Quality of English Language

Line 37: remove "s" from "keys".

Line 87: add "a" between "Utilizing" and "systematic"

Line 337: change "remind" to "reminder"

Author Response

(The authors gave the same response as above.)

Reviewer 3 Report

Comments and Suggestions for Authors

1) The Authors performed an interesting article entitled "Insomnia Medication Use by University Students: A Systematic Review". Overall, the strengths of the publication are the clinical interest of the topic and its impact on Public Health, and the number and quality of the bibliographic references (75). Additionally, I think certain aspects should be improved and commented on.

2) A suggestion: include in the abstract the main drugs used to treat insomnia?

3) Lines 56 - 65: the pharmacological treatment of insomnia should be improved, presenting examples of the main drugs, their therapeutic indications, and undesirable effects by pharmacological class.

4) Line 58: Melatonin agonists: please present examples of drugs.

5) The valerian should be mentioned as it is an herbal medicine indicated for relieving mild nervous tension and difficulty falling asleep.

6) Lines 58/59: Sedative antihistamines - please mention that they are 1st generation H1 antihistamines and present examples of drugs (for example: doxylamine).

7) Please address the use of benzodiazepines in greater detail, highlighting their undesirable effects and risks in university students (sedation, cognitive impairment (decrease memory, reasoning, and attention), physical and psychological dependence, pharmacological tolerance, etc.).

8) Do not use the term "pill" – please use "tablet" or "oral solid dosage form" or "capsule" (if appropriate).

9) Why didn't you present the drugs most used to treat insomnia in the various studies mentioned?

10) Lines 186-188: Four studies [34, 37, 43, 46] suggested that students who smoked reported significantly higher insomnia medication consumption compared to non-smokers. What is(are) the reason(s)? Is it because nicotine is a central nervous system stimulant? Please explain.

11) Lines 199 and 200: Another three studies found a significant association between drinking alcohol and using insomnia medication. Please indicate the consequences/interactions from a pharmacological point of view of taking simultaneously medications for insomnia (for example, doxylamine and benzodiazepines) and alcohol.

12) What role can physicians play in prescribing and advising insomnia medications for university students? I think this aspect should be explored in the article.

13) What role can pharmacists play as medicinal product specialists in dispensing and advising insomnia drugs for university students? I think this aspect should be explored in the article.

Author Response

(The authors gave the same response as above.)

Round 2

Reviewer 1 Report

Comments and Suggestions for Authors

The authors have answered the reviewer comments and the manuscript is much improved. However there remain some more minor comments requiring review by the authors for further changes in the text, which are summarised below:

- in the introduction, can the authors please specify in the text whether or not an existing review is published in this area, e.g. 'we could find no existing review'. The authors have now included reference 35 which is helpful, can the text be amended to refer to how bringing the literature together can help?

- In the introduction and/or methods, can the authors please make specific reference to the ways in which they could extract data on medication usage, specifically mentioning specific tools like PSQI. This helps the reader understand what data they collect and how they were useful for the research team. The additions to the results section about these tools are also welcome but some text earlier explaining how authors would deal with the data they found are needed.

- line 174: (such as gender, sleep performance and lifestyle factors).

- line 190: please amend as appropriate, e.g. 'two studies found a statistically significant difference...'

- can the authors please include a completed PRISMA checklist in their appendix.

- can the authors please specify in the methods text that MW completed data extraction from included studies, and RJ/DG independent checking of these (as per their response to reviewer documents). Can the authors please also describe the 'scope searching' in the methods (adding text in) for the reader, including what this entailed and who did it.

- can the authors please include in the methods text how they presented their prevalence and determinants data. If they did not manipulate any data and listed data as reported in the paper they should say so. They should also present in the text what was considered to be 'medication usage' by students, and later in the results how different studies defined this measure.

- can the authors please confirm in the paper methods text (a) how the search strategy was developed (who was involved, what process) e.g. whether a librarian supported it, and (b) of the studies included from reference list searching, whether these were indexed in the databases searched by the authors for the review. If they were indexed, the authors need to list in their limitations the possibility that some relevant studies may have been missed.

- tables 3-6. Some tables are formatted much bigger than others? Can the authors please amend the titles of these tables so it is clearer what the OR/P-values are representing, e.g. in the title it should state the relationship such as increasing or decreasing physical exercise activity and insomnia medication usage.

Comments on the Quality of English Language

No further comments.

Author Response

Thanks for your comments, and the major amendments are in the methods section.

Reviewer 3 Report

Comments and Suggestions for Authors

The Authors performed several explanations and alterations, and the manuscript is now suitable for publication.

Author Response

Thanks for your suggestions.